# Twenty-Four-Hour Movement Behaviors, Fitness, and Adiposity in Preschoolers: A Network Analysis

**Alyce Rodrigues Souza** [1], **Paulo Felipe Ribeiro Bandeira** [2,3,*], **Morgana Alves Correia da Silva** [2], **Glacithane Lins da Cunha** [2], **Daniel Fernandes Pereira** [2] and **Clarice Martins** [1,4,*]

1    Federal University of Paraíba, Cidade Universitária, João Pessoa 58000-000, Brazil
2    Federal University of Vale do São Francisco, Petrolina 56304-917, Brazil
3    Regional University of Cariri, Crato 63105-000, Brazil
4    Laboratory for Integrative and Translational Research in Population Health (ITR), Research Centre of Physical Activity, Health and Leisure, Faculty of Sports, University of Porto, 4500 Porto, Portugal
*    Correspondence: paulo.bandeira@urca.br (P.F.R.B.); claricemartinsufpb@gmail.com (C.M.)

**Abstract:** The present study aimed to verify the associations between compliance with the 24-h movement behavior recommendations, fitness, and adiposity markers in preschoolers, considering the non-linear nature of these associations. The sample was comprised of 253 preschoolers. Preschoolers were assessed for anthropometric data and wore an accelerometer for seven consecutive days. Screen time and sleep duration were parent-reported in a face-to-face interview. The PREFIT test battery was used to assess physical fitness components (lower-body strength, cardiorespiratory fitness, and speed/agility). Descriptive statistics were used to describe the variables, and a network analysis was conducted to assess the emerging pattern of associations between the variables. Preschoolers' greatest compliance with recommendations was observed for physical activity, while the lowest compliance was observed for the screen time recommendation. Among children aged three years, only 2.2% complied with all recommendations; only 1.0% of the four-year-olds and 1.3% of the five-year-olds complied with all recommendations. The results of the network analysis and centrality measures emphasized that cardiorespiratory fitness (CRF) and compliance with movement behavior recommendations were the most critical variables to address in preschoolers, reinforcing the importance of intervention programs focused on intense activities.

**Keywords:** children; obesity; complexity

## 1. Introduction

Childhood adiposity constitutes a global health problem [1]. Between 1980 and 2013, the prevalence of obesity among children increased by 47.1% [2], and according to the World Health Organization (WHO) [3], by 2025, the number of children with overweight and obesity can reach 75 million. This evidence becomes even more relevant when considering that the obesity epidemic has direct implications for public health; in addition, it leads to a considerable increase in health costs [4].

Obesity has been previously linked to at least 100 associated factors of different natures [5], including movement behaviors (physical activity (PA), sedentary behavior, and sleep). When analyzing the association between each movement behavior and adiposity, evidence has indicated a negative association between PA time and adiposity [6,7]. Previous studies have also shown that sleep duration in children is inversely associated with adiposity [8–10], while short sleep duration is associated with a higher body mass index (BMI) and waist circumference (WC) [11]. Moreover, regardless of PA levels, sedentary behavior, especially the time exposed to screens, is associated with an increased risk for adiposity and low physical fitness levels [12]. Physical fitness indicates the ability to engage in daily physical activity (PA) without excessive fatigue, respond to environmental demands, and maintain and improve health [13]. Physical fitness components play a protective role

in physical, emotional, mental, and social health in childhood [14], as cardiorespiratory fitness substantially benefits cardiovascular health [15]; strength training may improve body composition, with reductions in BMI [16,17]; and agility explains the variance of moderate-to-vigorous physical activity (MVPA) levels in childhood [18]. Indeed, physical fitness is only partly genetically determined and is greatly influenced by several other factors of different natures, such as movement behaviors [14]. These factors are non-linear and dynamically interconnected, which gives them the characteristics of a complex system. In this type of system, small changes in a single component can result in important non-deterministic patterns throughout the network of associations between the interconnected variables that comprise the system [19].

The WHO recommends that three- and four-year-old children accumulate at least 180 min of PA daily, of which at least 60 min should be MVPA; they should not spend more than 1 h on recreational screens; and they should obtain good-quality sleep for between 10 and 13 h a day. For five-year-olds, recent recommendations established that a healthy 24-h day should include at least 60 min of daily MVPA [1]. Tremblay et al. [20] stated that besides PA, children at this age should limit sedentary screen time to less than 2 h and sleep between 9 and 11 h per day. Nonetheless, data from 23 countries have shown that the majority of young children do not comply with the three recommended behaviors of the 24-h movement guidelines, and one in five do not comply with any of the three recommendations [21–23]. It is notable that Brazilian children seem to have one of the lowest levels of adherence among several countries [23]. Furthermore, compliance with the 24-h movement recommendations independently or in combination was significantly associated with a lower BMI z-score [24].

Considering that non-compliance with movement behavior recommendations, which is associated with low physical fitness, is a risk factor for adiposity and that behaviors are established in early childhood and persist throughout life [23], it is necessary to investigate the associations between these variables in a critical phase of adiposity development, such as early childhood. Considering that the interconnections between these variables are understood as complex non-linear systems, they could be better explored as a network of variables that forms emerging patterns, enabling the identification of more sensitive variables to maintain or modify the entire system. This perspective allows the evaluation of the role of each variable within the system. By calculating the expected influence index, it is possible to determine the variables more sensitive to changes resulting from interventions [19], which can influence the entire system. Thus, exploring the associations between compliance with movement behavior recommendations and modifiable risk factors for the emergence of obesity, such as physical fitness and adiposity markers, from the approach of complex systems will provide a better understanding of the dynamical interrelationships between these variables, and it may also support the development of actions to promote healthy lifestyles in preschoolers. In fact, there is no study that the authors are aware of that has explored these relationships from a network approach or hypothesized whether there is a non-linear relationship between these variables, and if so, how they are related from the perspective of complexity. Thus, this study aimed to verify the associations between compliance with the 24-h movement behaviors recommendations, fitness, and adiposity markers in preschoolers, considering the non-linear nature of these associations.

## 2. Materials and Methods

This cross-sectional study used baseline data from the "Movement´s Cool Project", which aimed to explore the associations between movement behaviors and health outcomes in low-income preschoolers. The main project was approved by the (removed for blind review).

### 2.1. Setting and Population Characteristics

Preschoolers aged 3 to 5 years of both sexes registered in early education childhood centers (EECCs) in João Pessoa were eligible and invited to participate. João Pessoa is a

large seaside city in the northeast of Brazil. We conveniently selected two EECCs in the coastal and central districts to be included in the study. All children aged 3 to 5 years with typical neurological development who were attending the two EECCs (256) were invited for assessments. Of those, 253 presented signed informed consent, and three did not complete the study´s protocol. Thus, the final sample was comprised of 253 preschoolers. The majority (62.5%) of mothers and fathers were unemployed. Over 45% of the mothers and 54% of the fathers had finished 9th grade or lower. The Human Development Index (HDI) of the EECCs' area ranges from 0.4 to 0.5.

## 2.2. Procedures

Assessments were conducted during a four-month period (November/December 2019 and February/March 2020). All the preschool staff and the preschoolers' parents were informed about the research protocols and procedures in meetings with the project coordinator (one session in each school) and agreed to participate. Trained physical education teachers and graduate students conducted the assessments.

The school administration provided the children's ages, birth dates, and parent contacts. Parents were invited to a meeting at the preschool and were interviewed individually. Social information and screen and sleep time were collected during this interview, and parents were also informed about the accelerometer used. Anthropometric data and resting heart rate beat-to-beat data were assessed at the preschools.

## 2.3. Measurements

### 2.3.1. Anthropometric Measures

Height (cm) and body mass (kg) were assessed using a Holtain stadiometer and a weighting scale (Seca 708, Hamburg, Germany) while the participant was lightly dressed and barefoot. Two measures were taken, and if they differed, the average value was adopted. BMI was calculated by dividing body weight by the squared height in meters ($kg/m^2$) [25], and BMI z-scores were calculated according to the WHO cut-offs [25].

### 2.3.2. Physical Activity

PA was objectively assessed using accelerometry (Actigraph WGT3-X, Pensacola, FL, USA, which is validated for measuring PA in preschoolers [26]. The preschool teachers received verbal and written instructions for the accelerometer's correct use. The teachers were instructed to register an activity diary of wear and non-wear time. The device initialization, data reduction, and analysis were performed using ActiLife software (Version 6.13.3).

Participants were advised to wear the right hip accelerometer for seven consecutive days (Wednesday morning to Tuesday afternoon). The children were allowed to remove the device during water-based activities and while sleeping (at night). During preschool, the teachers removed the accelerometers around 11 am for the children's baths and fastened them adequately afterward. During the week, parents received messages on their cellphones to remember to keep the accelerometer on their children.

Accelerometers were analyzed as ActiGraph counts considering the vector magnitude and a 15-s epoch length [27]. Periods of $\geq$20 min of consecutive zero counts were defined as non-wear time and removed from the analysis. The first day of data was omitted from the analysis to avoid subject reactivity [28]. Only days with a minimum of 8 h of wear time were considered valid. The mean wear time was 10.9 h (SD $\pm$ 1.4 h of wear time between children). The Butte et al. [29] cut points for light (820 to 3907 counts), moderate (3908 to 6111 counts), and vigorous ($\geq$6112 counts) PA intensity were used.

### 2.3.3. Sleep Time

Parents reported the children's usual daily sleep hours. This approach has been validated against estimates from sleep logs of objective actigraphy in young children [24]. Parents were asked to recall the total average hours their child slept as follows: "On weekdays, how many hours of sleep does your child usually have during the night?" and "On weekend days, how

many hours of sleep does your child usually have during the night?". Questions were asked separately for weekdays and weekend days and reunited for the analyses. Overall sleep hours were calculated as follows: ((Sleep on weekdays $\times$ 5) + (Sleep on weekend days $\times$ 2))/7.

### 2.3.4. Screen Time

Parents were also asked to recall the total average duration their child watched TV and used the computer, smartphones, and video games. The questions were asked separately for weekdays and weekend days and reunited for the analyses (Cronbach's $\alpha$ = 0.87). For screen time, the questions were: "How many hours during a week day does your child usually watch TV, use computer, smartphones or electronics games?" and "How many hours during a weekend day does your child usually watch TV, use computer, smartphones or electronics game?". Then, the same procedure used for sleep hours was applied.

### 2.3.5. Physical Fitness

Reliable and feasible assessments of three health-related fitness tests from the PREFIT Battery [28] were assessed, as follows:

1.  Cardiorespiratory fitness (CRF) was measured using the PREFIT 20-m shuttle run test [28]. Participants completed the PREFIT 20-m shuttle run keeping in time with an audible "bleep" signal. The frequency of the sound signals was increased every minute by 0.5 km/h, increasing the intensity of the test, and the children were encouraged to run to exhaustion. Some adaptations of the original test were made by decreasing the initial speed (i.e., 6.5km/h instead of the original 8.5 km/h). Evidence for the acceptable reliability and validity of the PREFIT 20-m shuttle run test for preschoolers was previously provided [27].
2.  Speed–agility (shuttle run test 4 $\times$ 10 m) consisted of running and turning as fast as possible between two parallel lines (10 m apart), covering a distance of 40 m. The best of two attempts was recorded (seconds). This test had a good correlation in boys and girls (r = 0.86) [27].
3.  Lower-body muscular strength was assessed by the standing long jump. From a parallel standing position and with the arms hanging loose at the sides, participants were instructed to jump twice as far as possible in the horizontal direction and land on both feet. The test score (the best of two trials) was the distance in centimeters, measured from the starting line to the point where the back of the heel landed on the floor, as previously proposed [27].

### 2.4. Statistical Analysis

Descriptive statistics are described as means and standard deviation and were calculated for the assessed variables, and one-way ANOVA with Tukey's post hoc test was performed to compare the differences between means by age. The prevalence of compliance by age for each of the recommendations was calculated.

Considering the importance of the assessed variables for an adequate emergent pattern, a non-linear machine learning approach entitled Network Analysis was used to explore the relationships between compliance with movement behaviors, physical fitness, and adiposity (BMI z-score and WC) according to age. This technique aims to establish relationships through multiple interactions between variables from graphical representations [30]. Concerning the cross-sectional nature of this study, an undirected weighted network analysis was used to estimate the relationship between nodes (assessed variables) from a correlation matrix; when transformed, they were represented by positive or negative edges, which were the relationships between the different nodes, but with no arrowheads to indicate the direction of effect.

The Fruchterman–Reingold algorithm was applied so that the data were presented in the relative space in which variables with stronger associations remained together and the less strongly associated variables were repelled from each other [31]. The pairwise Markov random field model was used to improve the accuracy of the partial correlation

network, which was estimated from L1 regularized neighborhood regression. The least absolute contraction and selection operator was used to obtain regularization and make the model less sparse [32]. The EBIC parameter was adjusted to 0.25 to create a network with greater parsimony and specificity [33]. The qgraph package of the R studio (free version) program was used to estimate and visualize the graph [33]. Regularized algorithms of the selection operator and minimum absolute reduction (LASSO) were used to obtain the precision matrix, which, when standardized, represented the associations between the network variables. The thickness and color intensity of the lines represent the magnitude of the associations. The blue lines represent positive associations, and the red lines represent negative ones.

Finally, the centrality index's expected influence was calculated. The expected influence indicates the importance of a node for the structure and function of the network. This centrality measure consists of the sum of all possible edge weights that connect one node to another and is used to assess the nature and strength of a variable's cumulative influence within the network and thus the role it is expected to play in the activation, persistence, and remission of the network [34]. A positive expected influence means that the influence of that specific node in the network tends to increase for the acquisition of an adequate network pattern [34].

## 3. Results

A total of 253 preschoolers ($4.44 \pm 0.76$ years old) were assessed. Significant differences between ages were seen for body weight and body height (Table 1).

**Table 1.** Carerísticas da amostra, estratificada por idade.

| | 3 | | 4 | | 5 | |
|---|---|---|---|---|---|---|
| | **M ± SD** **(N = 89)** | **p₁** | **M ± SD** **(N = 92)** | **p₂** | **M ± SD** **(N = 72)** | **p₃** |
| **Movement behaviors** | | | | | | |
| LPA (min/day) | $212.88 \pm 49$ | 0.075 | $213.55 \pm 53$ | 0.930 | $216.47 \pm 52$ | 0.893 |
| MVPA (min/day) | $54.57 \pm 19$ | 0.443 | $58.67 \pm 18$ | 0.442 | $63.01 \pm 24$ | 0.095 |
| TPA (min/day) | $267.45 \pm 57$ | 0.865 | $272.22 \pm 62$ | 0.740 | $279.49 \pm 65$ | 0.492 |
| Screen (min/day) | $169.75 \pm 111$ | 0.702 | $158.47 \pm 83$ | 0.551 | $173.99 \pm 83$ | 0.478 |
| Sleep (min/day) | $565.11 \pm 68$ | 0.791 | $558.55 \pm 67$ | 0.171 | $577.74 \pm 68$ | 0.193 |
| **Anthropometrics** | | | | | | |
| Weight (kg) | $16.37 \pm 2$ * | <0.001 | $18.00 \pm 3$ | <.001 | $19.66 \pm 3$ | <0.001 |
| Height (cm) | $101.13 \pm 6$ * | <0.001 | $106.23 \pm 5$ | <.001 | $112.17 \pm 6$ | <0.001 |
| BMI (kg/m²) | $15.96 \pm 1$ | 0.939 | $15.88 \pm 1$ | 0.475 | $15.59 \pm 1$ | 0.329 |
| WC (cm) | $52.15 \pm 3$ | 0.604 | $51.58 \pm 3$ | 0.935 | $50.72 \pm 3$ | 0.564 |
| **Physical fitness** | | | | | | |
| CRF (laps) | $19.45 \pm 10$ | 0.641 | $16.33 \pm 9$ | 0.060 | $17.50 \pm 8$ | 0.218 |
| Lower-body strength (cm) | $76.07 \pm 22$ | 0.320 | $71.3 \pm 24$ | 0.223 | $75.13 \pm 23$ | 0.127 |
| Speed–agility (s) | $10.90 \pm 4$ | 0.299 | $9.21 \pm 3$ | 0.993 | $11.54 \pm 4$ | 0.076 |

p₁: Significance of the difference between the ages of 3 and 4 years; p₂: significance of the difference between the ages of 4 and 5 years; p₃: significance of the difference between the three ages. One-way ANOVA with Tukey's post hoc; Note: BMI: body mass index; AFL: light physical activity; MVPA: moderate-to-vigorous physical activity. * significative *p* value.

PA was the behavior with the greatest compliance in preschoolers, while the lowest compliance was seen for the screen time recommendation. According to age, compliance with the three recommendations simultaneously was 2.2, 1.0, and 1.3%, for 3-, 4-, and 5-year-olds, respectively. (Figure 1).

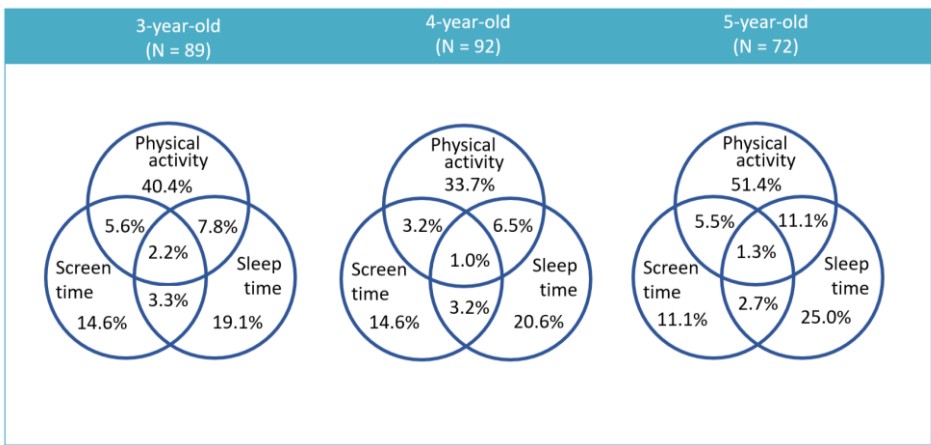

**Figure 1.** Venn diagram of preschoolers' compliance with movement behavior recommendations.

The network analysis highlighted the emergent patterns of the interrelationships between all the variables presented in the network. For the 3-year-old preschoolers, the emerging network (Figure 2) showed that compliance with PA and sleep recommendations were negatively associated with sex and positively associated with CRF. At 4 years old, the emergent pattern highlighted a negative association between compliance with PA recommendations and sex and a positive association between CRF and speed–agility. Moreover, compliance with the screen time recommendation showed a positive association with adherence to the sleep duration recommendation. Finally, adherence to the sleep duration recommendations showed a positive association with CRF. At 5 years old, the emergent pattern indicated a positive association between compliance with the PA recommendations and CRF and lower limb strength. Additionally, sex was negatively associated with physical fitness components.

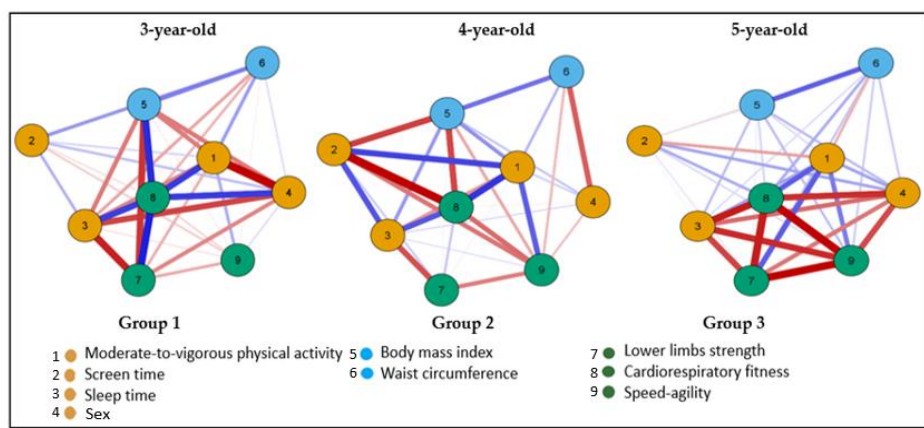

**Figure 2.** Preschoolers' emergent patterns, according to age.

When analyzing the centrality indexes for each age group, it was observed that for preschoolers aged 3 years, CRF presented the highest expected influence value (2.169). For the 4- and 5-year-old preschoolers, compliance with the PA recommendation was the variable with the highest value (2.222 and 1.309, respectively). (Table 2).

**Table 2.** Expected influence values of variables.

| | Expected Influence | | |
|---|---|---|---|
| **Variables** | **3** | **4** | **5** |
| Physical activity | −0.213 | 2.222 | 1.310 |
| Screen | 0.596 | −0.436 | 0.577 |
| Sleep | −0.995 | 0.774 | −0.911 |
| BMI z-score | −0.498 | −0.084 | 1.260 |
| WC | 0.616 | 0.058 | 0.936 |
| Lower body strength | −0.910 | −0.495 | −0.841 |
| CRF | 2.169 | −0.481 | −0.846 |
| Speed–agility | −0.104 | −1.315 | −0.997 |

Note: BMI = body mass index; WC = waist circumference; CRF = cardiorespiratory fitness.

## 4. Discussion

While previous studies have investigated compliance with the 24-h movement recommendations in preschoolers [23,24], this study offers unique insight into compliance with these recommendations and modifiable risk factors for obesity, considering the non-linear nature of these associations. Our main results indicated low compliance with the recommendations for the three movement behaviors simultaneously and highlighted a lower prevalence of compliance compared with previous results about the associations between movement behaviors and adiposity indicators. For instance, previous results have shown a compliance prevalence of 93.1% for Australian children and 61.8% for Canadian children [35,36]. Nonetheless, it is important to note that the present study reported the prevalence according to age, allowing us to better identify age-related variability and the best period to implement intervention strategies. Moreover, the current results originated from socially vulnerable regions, and the participants were low-income preschoolers, who engage in less structured PA [37], show a greater level of sedentary behaviors [38,39], and belong to families whose levels of sedentary behavior, especially screen time, are documented to happen earlier in life [40].

Although it is recognized that young children should be encouraged to play freely, it is also important to establish that fitness is an important mediator for a positive relationship between PA, motor competence, and consequent healthy weight status. The current results highlight different associations between movement behaviors, fitness components, and adiposity according to age [38]. At age 3, compliance with the PA recommendations was negatively associated with compliance with sleep duration, sex, BMI, and lower limb strength, and it was positively associated with screen time. Although it is somewhat controversial, it is possible to argue that at 3 years of age, these relationships are not well established yet [41]. While it is known that screen time in the first years of life presents a greater relationship with involvement in PA throughout the day, there is also evidence that increased screen time in parents is positively associated with more screen time in children [41,42], suggesting that other factors that were not considered in this study could determine the observed relationships.

To reinforce the abovementioned, at 4 years old, the associations observed became stronger once associations that were previously negative became positive from that age onward. Our data indicate that adherence to the PA recommendations was positively associated with sex, BMI, WC, lower limb strength, CRF, and speed–agility. Despite these positive results, adherence to the PA recommendation also showed a positive association with screen time, which could be credited to compensatory sedentary behavior, and a negative association with sleep duration. This negative association between PA and sleep duration at young ages has been previously shown. In fact, at young ages, children´s sleep patterns are not well established, and more PA, especially before bedtime, could negatively impact children´s sleep [40].

One explanation for this finding is that bedtime can directly interfere with adiposity measures, which is in line with the study by Xiu et al. [11], who concluded that more

frequent exposure to late sleep was associated with greater increases in adiposity measures in children aged 2 to 6 years, especially when parents were obese.

At 5 years old, compliance with the PA recommendation was positively associated with sleep duration, sex, BMI, lower limbs strength, CRF, and speed–agility. Moreover, PA compliance was negatively associated with compliance with screen recommendations and WC. These results reinforce the importance of an early-onset intervention for childhood obesity.

The main strength of the current study was the novel approach used to explore the associations between compliance with movement behavior recommendations, fitness, and adiposity markers in preschool children, accounting for their intrinsic non-linear interrelationship, as seen in a real-life context. This approach allows the evaluation of the interactions between variables as a complex system based on measures of centrality [43]. In addition, keeping variables that have small effects on the complex system is also important, considering that a small effect can be responsible for important changes in the entire network [44]. However, notwithstanding the novelty of the present study, some limitations should be highlighted. Ensuring that children wear sensors at night is a real ecological barrier to objectively assessing sleep duration at such young ages. The variability in the fitness assessments in this age group is also an important factor that should be recognized; this diminished our ability to gather detailed insight into certain ages. Moreover, other possible correlates of movement behaviors, fitness, and adiposity markers, such as preschoolers´ nutritional behaviors, could be explored in future studies. Finally, although the observed results reinforce the importance of early intervention in preschoolers to avoid the aggravation of modifiable risk factors for the development of obesity, concerning its cross-sectional nature, we advocate longitudinal designs that lay out the developmental course of the observed associations and allow further exploration of the stability and the prediction of changes in the investigated networks.

## 5. Conclusions

This study emphasized CRF and compliance with PA recommendations as the most critical variables to address in preschoolers, reinforcing the importance of interventions based on intense activities even in early childhood.

**Author Contributions:** Conceptualization, A.R.S. and C.M.; methodology, A.R.S., P.F.R.B., M.A.C.d.S., D.F.P., G.L.d.C. and C.M.; software, A.R.S., P.F.R.B., M.A.C.d.S., G.L.d.C. and C.M.; vali-dation, A.R.S., P.F.R.B. and M.A.C.d.S.; formal analysis, P.F.R.B. and C.M.; investigation, A.R.S., P.F.R.B., M.A.C.d.S., G.L.d.C. and C.M.; resources, A.R.S. and C.M.; data curation, A.R.S. and C.M.; writing—original draft preparation, A.R.S. and C.M.; writing—review and editing, P.F.R.B. and C.M.; visualization, A.R.S., P.F.R.B., M.A.C.d.S., G.L.d.C. and C.M.; supervision, P.F.R.B. and C.M.; project administration, P.F.R.B. and A.R.S.; funding acquisition, A.R.S. and C.M. All authors have read and agreed to the published version of the manuscript.

**Funding:** BPI-FUNCAP Research Productivity Grant 04-2022.

**Institutional Review Board Statement:** The study was conducted in accordance with the Declaration of Helsinki and approved by the Research Ethics Committee of the Health Science Center of the Federal University of Paraiba (protocol no. 2.727.698) and by the Education Board of João Pessoa city for studies involving humans.

**Informed Consent Statement:** Not applicable.

**Data Availability Statement:** Not applicable.

**Conflicts of Interest:** The authors declare no conflict of interest.

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
