# Peer review of "Twenty-Four-Hour Movement Behaviors, Fitness, and Adiposity in Preschoolers: A Network Analysis"

_2673-4168, doi:10.3390/obesities3010004_

Round 1
Reviewer 1 Report
To the authors,
Thank you for the opportunity to review this paper, which is entitled “24-Hour Movement Behaviours, Fitness and Adiposity in Preschoolers: A Network Analysis.” It is great to see associations between movement behaviors, physical fitness, and weight status in preschoolers, and I acknowledge the time and effort that went into this paper. However, there are some concerns about key elements of a scientific paper. As such, I could not recommend the acceptance of this paper, but I hope that my feedback below may be helpful in the future.
General feedback:
This study examined the associations between movement behaviors, physical fitness, and weight status in preschoolers. The authors showed visually non-linear associations by network links of factors for behavioral and physiological aspects of children. However, the reviewer could not understand this study's research question, hypothesis, and specific purposes. Thus, if the paper publishes, readers could see the associations visually, but it seems that most readers would not assess quantitively how the findings are different from previous findings.
Specific comment:
1. Rationale: The authors should improve the rationales for this paper. The reviewer could not understand clearly why the authors needed to use network analysis for examining associations between these factors. The authors should add the rationale and necessity of using network analysis in this study.
2. Analysis: Why did not the authors perform and show common association analyses such as multiple regression analysis and path analysis? The reviewer thinks that data from such analyses could help the understandability of the results from the network analysis. The reviewer recommends the authors rethink and revise the presentation of the study results from the viewpoint of understandability.
3. Physical fitness of kids: At the ages of children used in this study, children can play physically, but even though their voluntarily maximal abilities were measured, it would be difficult to check whether it was the true value of their maximal abilities since it is highly possible that the children do not understand tasks of fitness tests. Elementary school ages of children can obtain reliable data on physical fitness, due to they are cognitively and physically developed. In this study, how do the authors address the above-mentioned points? Please add the explanations for that and the reliability levels of the fitness data.
Specific points:
4. Abstract: Please check participant numbers. In the abstract, the number of participants shows as “263” but in the methods as “253”. Which is correct?
5. Abstract and Introduction: Please revise the sentence of the study purpose “aimed to analysis…” The word “analysis” is an unclear word, please use a specific word.
6. Introduction: Please change word usage: from “obese children” to “children with obesity” throughout the manuscript.
7. Introduction: The reviewer could not understand what the authors show the mean of the third paragraph. The reviewer appears the paragraph just shows previous results, not link to the research question. Please revise the paragraph easy to get the author's thoughts shortly.
8. Please add the hypothesis of the study and the necessity for using network analysis.
The reviewer finds out many grammatical mistakes in English in the manuscript. The reviewer recommends the manuscript receive English proof-leading by a native English speaker. I hope these comments will be helpful.
Author Response
We wish to thank the reviewers again for their time and expertise in another review of our manuscript. We hope we have address the remaining concerns raised by the two reviewers which are provided below.
|
Reviewer 1 |
|
|
This study examined the associations between movement behaviors, physical fitness, and weight status in preschoolers. The authors showed visually non-linear associations by network links of factors for behavioral and physiological aspects of children. However, the reviewer could not understand this study's research question, hypothesis, and specific purposes. Thus, if the paper publishes, readers could see the associations visually, but it seems that most readers would not assess quantitively how the findings are different from previous findings. Please add the hypothesis of the study and the necessity for using network analysis |
Thank you, were have addressed your thoughtful comments below. |
|
ABSTRACT |
|
|
Please check participant numbers. In the abstract, the number of participants shows as “263” but in the methods as “253”. Which is correct? |
The change was made as requested. The correct number is 253.
|
|
Line: 13. Please revise the sentence of the study purpose “aimed to analysis…” The word “analysis” is an unclear word, please use a specific word. |
Corrected. |
|
INTRODUCTION |
|
|
Line: 82. Please revise the sentence of the study purpose “aimed to analysis…” The word “analysis” is an unclear word, please use a specific word. |
Corrected. |
|
Please change word usage: from “obese children” to “children with obesity” throughout the manuscript. |
Modified. |
|
Line: 55 - 73. The reviewer could not understand what the authors show the mean of the third paragraph. The reviewer appears the paragraph just shows previous results, not link to the research question. Please revise the paragraph easy to get the author's thoughts shortly. |
Thanks for the suggestion. The change was made. “. In fact, there is no study that the authors are aware of that explored these relationships from a network approach, allowing hypotheses whether there is a non-linear relationship between these variables; and if so, how they are related, from the perspective of complexity”. |
|
RATIONALE |
|
|
The authors should improve the rationales for this paper. The reviewer could not understand clearly why the authors needed to use network analysis for examining associations between these factors. The authors should add the rationale and necessity of using network analysis in this study. |
We appreciate the considerations. The justification for the work was reviewed, including the importance of using network analysis. |
|
METHODS |
|
|
Analysis: Why did not the authors perform and show common association analyses such as multiple regression analysis and path analysis? The reviewer thinks that data from such analyses could help the understandability of the results from the network analysis. The reviewer recommends the authors rethink and revise the presentation of the study results from the viewpoint of understandability. |
We appreciate the considerations. Modifications were included |
|
Physical fitness of kids: At the ages of children used in this study, children can play physically, but even though their voluntarily maximal abilities were measured, it would be difficult to check whether it was the true value of their maximal abilities since it is highly possible that the children do not understand tasks of fitness tests. Elementary school ages of children can obtain reliable data on physical fitness, due to they are cognitively and physically developed. In this study, how do the authors address the above-mentioned points? Please add the explanations for that and the reliability levels of the fitness data. |
We appreciate the considerations. Modifications were included |
Reviewer 2 Report
this is an original since you studied the pre school children but the paper is not clear
ABSTRACT
The present study aimed to analyze the associations between compliance to the 24-hour movement behaviors recommendations, fitness, and adiposity markers in preschoolers, considering the non-linear nature of these associations. The sample was comprised by 263 preschoolers.. Hypothesis ? age ? The school administration provided all socio-demographic data (children's age, birth date, parent's contact, and address). (are there the factors ?) Social information and screen and sleep time were collected during this interview, and parents were also informed about the accelerometer used. Not necessary in the abstract Anthropometric data and resting heart rate beat-to-beat data were assessed at preschools. Not necessary in the abstract. The prefit test battery was used to assess physical fitness (which one ?). Descriptive statistics were used to describe the variables and a network analysis was conducted to assess the emerging pattern of associations between the variables. Not necessary it is evident Physical activity was observed as the behavior with greater adherence, while the lowest adherence was observed for the screen time recommendation. Explain it is nebulous Among children aged three years, only 2.2% met all recommendations and only 1.0% of the four-year-olds met all recommendations. In the group of children aged five years, only 1.3% met all recommendations. The results of the network analysis and centrality measures of this study emphasized CRF not defined and compliance with recommendations to the most critical variable intervein in preschoolers, reinforcing the importance of interventions based on intense activities.
What are the factors ? Age, and ??? socio-demographic data
METHODS
Measurements Anthropometric measures Height (cm) and body mass (kg) were assessed using a Holtain stadiometer, and weighting scale (Seca 708, Germany), while the participant was lightly dressed and barefoot. Two measures were taken, if they differed, the average value was adopted. BMI was calculated by dividing body weight with the squared height in meter
Adiposity % MG is necessary BMI is not pertinent
Then, the Anova One-Way with Tukey's Post-Hoc was performed to compare differences between means by age.
And the socio-demographic data?
We know that it is of importance in matter of obesity
Please explain the table 1 and correct it
Table 1. Características da amostra, estratificada por idade.
Please explain the link between your rational that is not clear and the figure 1
It seems that you apply a model without any rational and hypothesis
Author Response
We wish to thank the reviewers again for their time and expertise in another review of our manuscript. We hope we have address the remaining concerns raised by the two reviewers which are provided below.
|
this is an original since you studied the pre school children but the paper is not clear |
In addition to the fact that it was carried out with preschoolers, it was understood from the perspective of complexity, understanding that there is no study that the authors are aware that explored these relationships from a network approach, allowing hypothesize whether there is a non-linear relationship between these variables; and if so, how they are related, from the perspective of complexity
|
|
ABSTRACT |
|
|
Line: 15. The present study aimed to analyze the associations between compliance to the 24-hour movement behaviors recommendations, fitness, and adiposity markers in preschoolers, considering the non-linear nature of these associations. The sample was comprised by 263 preschoolers.. Hypothesis ? age ? |
We appreciate the considerations. Modifications were included. |
|
Line: 15 – 17. The school administration provided all socio-demographic data (children's age, birth date, parent's contact, and address) (are there the factors ?) |
Yes, the school board and those responsible provided all sociodemographic data |
|
Line: 17 – 18. Social information and screen and sleep time were collected during this interview, and parents were also informed about the accelerometer used. Not necessary in the abstract |
Correct |
|
Line: 18 – 19. Anthropometric data and resting heart rate beat-to-beat data were assessed at preschools. Not necessary in the abstract. |
Correct |
|
Line: 19 – 20. The prefit test battery was used to assess physical fitness (which one ?) |
prefit -> PREFIT The description of the test battery PREFIT can be found in "materials and methods", subtopic "Physical fitness". |
|
Line: 20 – 21. Descriptive statistics were used to describe the variables and a network analysis was conducted to assess the emerging pattern of associations between the variables. Not necessary it is evident |
Correct |
|
Line: 22 – 23. Physical activity was observed as the behavior with greater adherence, while the lowest adherence was observed for the screen time recommendation. Explain it is nebulous |
We appreciate the considerations. Modifications were included. “Preschoolers´ greater compliance with recommendations were seen for physical activity, while the lowest compliance was seen for the screen time recommendation”. |
|
Line: 25 – 28. The results of the network analysis and centrality measures of this study emphasized CRF not defined and compliance with recommendations to the most critical variable intervein in preschoolers, reinforcing the importance of interventions based on intense activities. What are the factors ? Age, and ??? socio-demographic data |
We appreciate the considerations. Was changed to: “The results of the network analysis and centrality measures emphasized CRF and compliance with movement behaviours recommendations as the most critical variables to intervein in preschoolers, reinforcing the importance of intervention programs focused on intense activities”. |
|
|
|
|
METHODS |
|
|
Line: 117 – 122. Anthropometric measures. Height (cm) and body mass (kg) were assessed - Adiposity % MG is necessary BMI is not pertinent |
Although BMI and circumference are correlated variables, they have different functions in the network, for example when we look at centrality measures. Furthermore, there was no multicollinearity, so we decided to keep both variables. |
|
Line: 183 – 184. Then, the Anova One-Way with Tukey's Post-Hoc was performed to compare differences between means by age. And the socio-demographic data? We know that it is of importance in matter of obesity |
Although it is an important consideration, sociodemographic variables were considered in the analysis of networks |
|
Please explain the table 1 and correct it |
Correct |
|
Please explain the link between your rational that is not clear and the figure 1. It seems that you apply a model without any rational and hypothesis |
Correct |
Round 2
Reviewer 2 Report
this paper has been improved congratulations